# Gut Microbiota Composition and Cardiovascular Disease: A Potential New Therapeutic Target?

**DOI:** 10.3390/ijms241511971

**Published:** 2023-07-26

**Authors:** Martina Belli, Lucy Barone, Susanna Longo, Francesca Romana Prandi, Dalgisio Lecis, Rocco Mollace, Davide Margonato, Saverio Muscoli, Domenico Sergi, Massimo Federici, Francesco Barillà

**Affiliations:** 1Division of Cardiology, Department of Systems Medicine, Tor Vergata University, 00133 Rome, Italy; martynebelli@gmail.com (M.B.);; 2Cardiovascular Imaging Unit, San Raffaele Scientific Institute, 20132 Milan, Italy; 3Department of Systems Medicine, Tor Vergata University, 00133 Rome, Italyrocco.mollace@gmail.com (R.M.);; 4Division of Cardiology, Mount Sinai Hospital, Icahn School of Medicine at Mount Sinai, 1 Gustave L. Levy Place, New York, NY 10029, USA; 5Cardiovascular Department, Humanitas Gavazzeni, 24125 Bergamo, Italy

**Keywords:** gut microbiota, cardiovascular diseases, trimethylamine N-Oxide, heart failure, atherosclerosis

## Abstract

A great deal of evidence has revealed an important link between gut microbiota and the heart. In particular, the gut microbiota plays a key role in the onset of cardiovascular (CV) disease, including heart failure (HF). In HF, splanchnic hypoperfusion causes intestinal ischemia resulting in the translocation of bacteria and their metabolites into the blood circulation. Among these metabolites, the most important is Trimethylamine N-Oxide (TMAO), which is responsible, through various mechanisms, for pathological processes in different organs and tissues. In this review, we summarise the complex interaction between gut microbiota and CV disease, particularly with respect to HF, and the possible strategies for influencing its composition and function. Finally, we highlight the potential role of TMAO as a novel prognostic marker and a new therapeutic target for HF.

## 1. Introduction 

The human gastrointestinal (GI) tract contains a rich and complex population of microorganisms that constitutes the gut microbiota [1]. 

Considering the symbiotic interaction between the gut microbiota and their host, we can consider the host as a “superorganism” [2], whose metabolism is dependent on a combination of both microbial and human physiological processes, which perform immune and metabolic functions [1,3].

Gut bacteria play a role in the digestive process and contribute to the absorption of many nutrients and metabolites, including essential amino acids, vitamins, lipids, and short-chain fatty acids (SCFAs) [4]. The gut microbiota has a critical immune function that promotes host resistance to infectious diseases [5]. 

Everyone possesses a specific gut microbiota profile. This profile comprises different bacterial species taxonomically classified according to phyla, classes, orders, families, genera, and species [6,7]. 

Several factors can alter the gut microbiota, including genotype, dietary habits, age [8,9], mode of birth [10,11], and the use of antibiotics [12,13]. Disruption of the gut microbiota is associated with several human diseases, including inflammatory bowel disease (IBD) [14,15], obesity, diabetes [15], allergies [16], autoimmune disease [17], CV disease [18,19], and hypertension [20]. Modifying its composition and diversity is a promising treatment option for these diseases. There are many ways to modulate the gut microbiota, including probiotics, prebiotics, and the transplantation of the faecal microbiota, which can induce beneficial changes in its structure and functions.

Recent evidence suggests that there is a link between the gut microbiota and CV diseases, including HF. Despite the progress in pharmacological and non-pharmacological therapies, the latter is still burdened by very high mortality with poor a prognosis.

This review aims to highlight the complex interplay between the microbiota, its metabolites, and CV disease’s development and/or progression, emphasizing HF and giving exciting directions for possible future research.

## 2. The Link between Gut Microbiota and the Heart

In the last decade, scientific awareness of the gut microbiota and its possible link to heart disease has grown. In fact, chemicals or processes associated with gut bacteria have been linked to a higher risk of HF, atherosclerosis, and major CV events such as heart attack and stroke [21].

While there is still much to learn about how the gut microbiota affects various disease risks, it is clear that an unhealthy gut negatively impacts health through inflammation, that is, the immune system’s response to an injury or foreign substance. 

Seventy percent of the body’s inflammatory cells are in gut-associated tissue [22]. So, gut inflammation involves substances, including chemicals produced by gut bacteria, that leave the gut and enter the bloodstream. This causes an inflammatory response wherever they enter the body. When the inflammation affects the blood vessels, these vessels lose their elasticity, endothelial wall function is impaired, and subsequent pathophysiological conditions lead to atherosclerotic disease [23].

Many studies have shown that unhealthy gut microbiota and its active metabolites are involved in the onset and progression of CV disease. Among these metabolites, although the corresponding pathophysiological mechanisms are not yet fully understood, TMAO is the main substance capable of triggering the inflammatory process that leads to the development of both metabolic and CV diseases. 

In a meta-analysis study that included 19 clinical trials, the associated authors found that increased levels of TMAO were associated with a 62% increased risk of suffering major CV events, such as heart attack and stroke, and a 63% increased risk of death from all causes. These results were homogeneous across study populations [24].

Another meta-analysis study that evaluated 17 trials showed that high blood levels of TMAO were associated with a 67% and 91% increased risk of major CV events and death from all causes, respectively. The risk of death increased proportionally with the rise in levels of TMAO and not only at very high levels of this metabolite [25].

It is important to note that, according to the available data, even with adjustment for traditional risk factors—considering differences in blood pressure, cholesterol, or triglycerides—an elevated level of TMAO may predict an increased risk of CV events. 

Another notable finding was reported in an experimental study using mice where animals with atherosclerosis were fed a beneficial type of bacteria that uses TMAO for food. The consumption of this kind of food resulted in lower levels of TMAO in their blood and reduced amounts of atherosclerotic plaque [26].

Two types of supplements are commonly recommended to support healthy gut microbiota: probiotics and prebiotics. Probiotics contain beneficial bacteria, while prebiotics contain substances that can promote the growth of beneficial bacteria. While there is evidence that some types of these supplements benefit gut health [27], it has not yet been clarified whether taking probiotics or prebiotics can reduce inflammation and CV disease in the long term [28].

## 3. Gut Microbiota and Atherosclerosis 

Infectious processes are associated with the development of CV disease and atherosclerosis. Recent research has identified microbial ecosystems in different parts of the human body that cause CV and metabolic disorders through multiple pathways [29]. First, local or systemic infections can trigger inflammation contributing to plaque development and/or destabilisation [30]. Second, cholesterol and lipid metabolism via the gut microbiota may lead to the development of atherosclerosis. Third, dietary components metabolised by the gut microbiota may influence atherosclerosis in different ways [29]. Considering that the number of resident bacterial species differs between individuals, microbiotas are comparable at higher taxonomic levels (e.g., phyla). For example, Bacteroidetes and Firmicutes, which account for more than 90% of all taxa in the human intestine, are the two phyla that are most abundant in the gut microbiota [31]. An altered gut microbiota is associated with coronary artery disease. Takumi Toya et al. showed that the gut microbiota’s composition is different in patients with advanced CAD compared with controls, confirming that the microbiota may be a therapeutic target for this disease [32]. Plaque formation may be caused by an extended infection or a disease of the vessel wall cells; the discovery of bacterial DNA in plaques supports this mechanism [33]. The microorganisms associated with atherosclerotic plaques are also found in other body parts, especially in the intestine. Among the infections caused by bacteria in atherosclerotic plaques, the gut microbiota affects CV disease by modulating host metabolism, including lipoprotein metabolism. Patients with atherosclerosis usually have an impairment of lipid metabolism, which can be modulated by the bacterial taxa of the gut microbiota [34]. Honghong Liu et al. confirmed that the gut microbiota impacts cholesterol homeostasis by modulating bile acids and that CAD-associated bacteria affect vessel stiffness [35]. Some microbiota species generate enzymes that facilitate the fermentation of indigestible carbohydrates into SCFAs [36]. These SCFAs have potential anti-inflammatory properties. Bacteria regulate intestinal permeability. Certain species may render the gut permeable, allowing microbe-associated metabolites to enter the circulatory system and elicit an inflammatory response, following which our body produces cytokines and other inflammatory mediators [37]. As mentioned, gastrointestinal bacteria can convert complex carbohydrates otherwise undigested by the host into SCFAs. SCFAs play a crucial role in the relationship between diet, the gut microbiota, and the stimulation, or inhibition, of inflammatory cascades downstream. The gut microbiota can convert primary bile acids into secondary bile acids. Changes in gut microbiota affect the varieties of secondary bile acids synthesised. Of particular relevance in this regard is the fact that bile acids cause inflammation by triggering the farnesoid X receptor (FXR) signalling networks in enterocytes and adipocytes [38]. Choline, carnitine, and betaine, found in red meat, fish, and other animal sources, are converted by microorganisms in the gut into trimethylamine (TMA), which is converted by hepatic flavin monooxygenases into TMAO. When present in high serum concentrations, TMAO has been associated with adverse effects, including endothelial dysfunction, thereby promoting vascular inflammation, atherosclerosis, and another CV disease, namely, HF [39]. 

## 4. The Gut Hypothesis of HF

Numerous data support the pathogenetic role of gut microbiota in the development of HF, so the “gut hypothesis of HF” has become a fascinating topic.

It is widely acknowledged that congestive HF (CHF) is a multisystemic disorder often associated with alterations in gut microbiota composition and function [40]. A well-balanced gut microbiota, an intact mucosa, and a normally functioning immune system are necessary to preserve the normal functionality of the gut barrier [41]. 

Several studies have shown that patients with CHF can be distinguished from healthy individuals via an impairment of gut microbiota composition. The guts of patients with CHF may present an intestinal overgrowth of pathogenic bacteria (adherent bacterial populations, such as Bacteroides and Fusobacterium) and fungal species (such as Candida), which can cause chronic intestinal wall inflammation that leads to malabsorption and decreased metabolic efficiency [42]. 

An decrease in cardiac output and an increase in sympathetic activity in CHF patients lead to a redistribution of the blood flow away from the splanchnic circulation. These changes result in reduced intestinal mucosal perfusion, intramucosal acidosis, mucosal oedema (thickened bowel wall), ischemia, epithelial dysfunction, and increased mucosal permeability in the small and large intestines [43]. 

Impairment of gut barrier function allows bacteria and their products to cross the gut barrier, a process known as bacterial translocation, thereby spreading to the mesenteric lymph nodes or entering the bloodstream and reaching more distant organs. Although it has been generally considered a pathologic process, this process is common during early life and may aid mucosal antigen sampling in the gut [44]. Three mechanisms have been suggested to be responsible for the pathogenesis of bacterial translocation: bacterial overgrowth, altered gut barrier function, and impaired host defences [41] (Figure 1). Bacterial translocation increases the number of bacteria, as well as the circulation of endotoxin (such as lipopolysaccharide, or LPS), that can activate monocytes and macrophages in order to release pro-inflammatory mediators [45] that participate in chronic systemic inflammation, which is a hallmark of CHF [46]. The increased levels of circulating cytokines further compromise the already dysfunctional gut barrier, thereby promoting a further increase in bacterial translocation. 

Niebauer et al. found that patients with CHF and a recent onset of peripheral oedema presented increased serum concentrations of endotoxin and cytokines. After diuretic treatment, it was observed that endotoxin levels (but not cytokine levels) normalised, suggesting that endotoxin may trigger immune activation and peripheral oedema [47]. During acute or decompensated HF, endotoxin levels were higher in hepatic veins than in the left ventricle or pulmonary artery, suggesting endotoxin translocation from the gut to the bloodstream [48]. In addition, selective gut decontamination (aerobic Gram-negative bacilli eradication) in patients with severe CHF reduced faecal endotoxin concentrations, monocyte CD14 expression, and intracellular pro-inflammatory cytokine production [49]. 

Bacterial translocation’s role in HF pathogenesis is still being actively researched. However, it is recognised that the gut microbiota and its interaction with the host significantly impact cardiovascular health. Strategies with the aim of modulating the gut microbiota may offer a potential therapeutic benefit in HF management.

## 5. TMAO, Inflammation and CVD

TMAO can directly affect the heart by causing myocardial hypertrophy, fibrosis, inflammation, and mitochondrial dysfunction, indirectly promoting renal fibrosis and dysfunction and platelet hyperreactivity [50,51]. Zhiye et al. demonstrated in in vitro and in vivo studies that TMAO treatment directly caused cardiac hypertrophy and fibrosis, as evidenced by increased cardiomyocyte size and high levels of hypertrophic markers including atrial natriuretic peptide (ANP) and beta-myosin heavy chain (β-MHC). In addition, cardiac hypertrophy can be markedly inhibited by reducing the synthesis of TMAO via antibiotic therapy [52]. The Smad3 signalling pathway was found to be activated in mice via the induction of transverse aortic constriction [53], and it would appear to play a key role in hypertrophy and fibrosis. In fact, its inhibition by SIS3 [52] markedly reduces cardiac remodelling. Moreover, it was found that an inhibitor of the synthesis of TMAO, namely, 3,3-dimethyl-1-butanol (DMB), prevents myocardial hypertrophy and fibrosis by regulating transforming growth factor-β1 (TGF-β1)/Smad3 and nuclear factor p65-κB (NF-κB) [54]. These findings confirm the role of TMAO in ventricular remodelling. The NF-kB signalling pathway is also responsible for inflammation in vascular smooth muscle cells. TMAO activates mitochondrial reactive oxygen species (mtROS) by inhibiting both sirtuin 3 (SIRT3) expression and superoxide dismutase 2 (SOD2) activity, subsequently activating the inflammasome (NLRP3), leading to the extension of inflammation to endothelial cells [55]. The ability to assay reactive oxygen species with sophisticated laboratory techniques such as the d-ROMs (Reactive-Oxygen-Metabolites-derived compounds test) and the BAP Test (Biological Antioxidant Potential) could be useful in identifying the inflammatory process early, so ROS could be another promising marker in this field besides TMAO. Furthermore, TMAO promotes myocardial inflammation by increasing TNF-α levels and decreasing IL-10 levels, leading to increased expression of pro-inflammatory cytokines and decreased expression of anti-inflammatory cytokines, but the relevant mechanisms are still not well understood and the pertinent receptor molecular has not yet been identified. Therefore, at present, only the downstream mechanism is known. [56]. The effect of TMAO on endothelial dysfunction is partly attributable to the activation of PKC/NF-kB, resulting in increased VCAM-1 expression and monocyte adhesion [57]. TMAO decreases energy metabolism and mitochondrial function by influencing pyruvate and fatty acid oxidation, which are involved in the tricarboxylic acid cycle [58]. How does TMAO cause cardiac damage indirectly? TMAO directly contributes to renal interstitial fibrosis and dysfunction, promoting sodium and water retention. The kidney plays a key role in TMAO excretion [59], so impaired renal function causes its accumulation along with the progression of cardiac injury. Finally, TMAO directly increases platelet hyperreactivity, which promotes platelet adhesion to collagen and the mobilization of cytoplasmic Ca^2+^, thus promoting a prothrombotic phenotype with an increased risk of myocardial infarction [60]. (Figure 2).

## 6. TMAO: A Prognostic Marker of HF

The first description of an association between increased TMAO levels and mortality risk among patients with HF dates back to 2014 [61]. In advanced CHF, a common problem is venous congestion deriving from right-sided HF. The increase in venous pressure and splanchnic congestion in relation to the gut is associated with deleterious effects. Environmental alterations in the gut predispose the host to colonisation with harmful TMAO-producing bacteria, and the increase in TMAO levels, as in a negative loop, leads to the further worsening of HF [62]. Increased TMAO levels have also been shown to be a strong predictor of mortality among patients with HF with a reduced ejection fraction (HFrEF), but not among those with a preserved ejection fraction (HFpEF), over a mean follow-up of 9.7 years [63]. 

In a meta-analysis by Xingxing Li et al., the results of 12 studies involving 13,425 participants from 2014 to 2021 demonstrated a strong association between elevated concentrations of TMAO and poor prognosis among patients with HF. Of the 12 studies included in this meta-analysis, 9 included patients with chronic HF. From this analysis, it was determined that high TMAO levels correlate with a greater risk of MACEs and an increase in all-cause mortality among patients with HF [64]. Several studies have reported that elevated TMAO levels are associated with poor prognoses for patients with HFrEF [61,65]. 

In a recent study including 1208 patients with chronic HF following a myocardial infarction, using Cox regression analysis, it was possible to demonstrate the association between plasma TMAO levels and cardiovascular outcomes. TMAO might be considered a predictor of MACE and could be used to improve risk stratification for patients with ischemic heart disease developing chronic HF. By investigating the association between HFpEF and TMAO, Zengxiang Dong et al. showed that the plasma levels of this microbial metabolite were 6.84 μmol per L in the HFpEF group and 1.63 μmol per L in the control group. This result showed that TMAO is significantly associated with HFpEF [66]. Moreover, pre-clinical studies have shown that in a mouse model of pressure-overload-induced HF, the dietary supplementation of TMAO aggravates myocardial fibrosis and left ventricular remodelling and dysfunction. Elevated levels of TMAO, therefore, could be associated with the worsening of left ventricular diastolic function in addition to an increased left atrial volume and thus impaired tissue mechanics, the latter of which is a key feature of HFpEF [53]. 

However, the exact role of TMAO with respect to HFpEF is unclear. TMAO is a marker of HF as serum levels of TMAO are higher among patients with HF than those without HF, but, importantly, this does not discriminate HFpEF from HFrEF. Thus, TMAO cannot be considered a unique marker of HFpEF but can help identify high-risk patients when associated with other markers [67]. Acute HF is another important HF setting, for which TMAO is essential as a prognostic marker: in a study by Suzuki et al., the authors reported that the detection of elevated TMAO levels in patients admitted to the hospital for acute HF correlated with an increase in rehospitalization at 1 year and mortality. Moreover, high TMAO further strengthened the prediction of death or HF at 1 year when combined with another important HF marker, i.e., NT-proBNP. Patients with elevated levels of both markers were at the highest risk of death or rehospitalization for HF at 1 year [68]. In two different studies, Kinugasa et al. and Salzano et al. showed that elevated TMAO levels at discharge are associated with an increased risk of post-discharge cardiac events among patients hospitalized with acute HF and HFpEF, especially among those with poor nutritional statuses [69,70]. 

Since the effect of TMAO on prognosis may interact with a patient’s nutritional status, the prognostic effect of TMAO may depend on the phenotype of HFpEF. Furthermore, it has been shown that racial and geographic differences could affect TMAO values and their effects on prognosis in HF [71,72].

We can conclude that TMAO plays an important prognostic role in HF and all its settings, but further studies are needed to validate its use in common clinical practice.

## 7. Gut Microbiota Analysis in Clinical Practice

In the last decade, a strong association between changes in microbiota composition and several host diseases has been found [73]. In particular, dysbiosis plays an important pathophysiological role in cardio-metabolic diseases [74]. 

Advances in DNA-sequencing technologies have aided the study of the microbiota and its contribution to cardiometabolic risk [75]. Meta-omics techniques characterise microbial communities, allowing us to perceive their influence on human physiology beyond simply the profiling of taxonomic compositions [76,77]. They are high-throughput-sequencing (HTS) techniques performed at the microbial, DNA, and mRNA levels based on available samples and research goals [78]. The potential biological samples can be various, but stool samples are the most used [79].

Sequencing and amplification of the marker gene (amplicon) is performed using a PCR primer specific for a particular hypervariable region of a microbial gene [73]. This technique uses the 16S ribosomal ribonucleic acid (rDNA) gene for prokaryotes and the 18S rDNA gene and internal transcribed spacers for eukaryotes, wherein the conserved and variable regions exist [78,80]. Therefore, phylogenetic or taxonomic profiles are produced, using large public datasets for comparison [76]. Amplicon sequencing is the most used, fastest, and cheapest technique for samples with low biomass or host DNA contamination. Nevertheless, it provides low-resolution taxonomic information (only at the genus level) without information on the functional potential of the microbiota, which can only be speculated [76,77]. Furthermore, bias may occur during amplification because this process is sensitive to the specific primers and the number of PCR cycles chosen [75]. Therefore, this technique is often used in pioneering research [78].

Metagenomics uses short reads of DNA sequences and captures all DNA in a sample, including viral and eukaryotic DNA. Reference libraries and datasets provide more detailed genomic information and taxonomic resolution at the species or strain level [75]. Furthermore, metagenomics depicts a functional profile at the genetic level through a bioinformatic analysis of metagenomic sequences [76,78]. Metagenomics is useful for profiling the entire genetic repertoire of a study group and in viral ecology where specific marker genes to amplify are unavailable. This technique can also reveal the interactions and mechanisms between the gut microbiota and disorders [80]. However, it is expensive and time consuming, and it needs a much larger average genome size than 16S rDNA analysis and quality control protocols to negate the contamination of human DNA [76].

In meta-transcriptomics, gene expression is analysed through the reverse transcription of RNA isolated from a sample [76]. This provides information on the microbiota’s active functional output, discriminating metabolically active microbes from dormant or dead ones and extracellular DNA [75]. This technique also evaluates changes in microbial gene expression over time [76,77] and in response to external stimuli [75]. However, the associated sample collection, storage, and preparation steps are difficult to manage [76]; contamination from the host mRNA and rRNA must be removed; and the resulting data are biased toward organisms with high transcription rates [75].

Metaproteomics can identify and quantify expressed proteins in a sample using higher-resolution mass spectrometry and quantitative proteomics techniques. Furthermore, this technique can link proteins to the microorganisms that encode them using databases and bioinformatics tools [76]. The associated limitation is the lack of standardization of this method [77]. In metabolomics, metabolite levels and fluxes are measured using mass spectroscopy and nuclear magnetic resonance [76]. This method provides a readout of the host–microbiota interface using a range of known metabolites (i.e., a targeted search, with higher sensitivity and yielding semi-absolute quantifications) or covering as many metabolites as possible (i.e., an untargeted search, the results of which are harder to interpret) [77]. Incomplete databases and the difficulty of distinguishing metabolites from microbiota and material of host origin limit this technique’s application [76].

Multi-omics approaches integrating data from different HTS methods provide a further opportunity to understand the composition and function of microbial communities [76] (Figure 3).

## 8. Therapeutic Perspectives for CVD and HF

Data have been collected demonstrating that several therapeutic options based on the re-balancing of dysbiosis involving gut and liver microbiota may be helpful in developing better intervention strategies for counteracting metabolic disorders, including atherosclerotic vascular disease states (ATVD). On the other hand, due to the huge correlation existing between microbiota modifications and the development of atherosclerosis and hypercholesterolemia, the effects of drugs commonly used for the treatment of these conditions may be impacted [81]. Recently, gut microbiota alterations have also been shown to be involved in the pathogenesis and pathophysiology of HF and its comorbidities [80,82,83]. HF still represents a major clinical problem worldwide regarding mortality, hospitalization, and public health [84]. In the last decade, many advances have been made regarding the clinical approach to HF [85,86]. However, there are still several unmet needs, especially concerning the treatment of the early stages of the disease, mostly in the area of HF with HFpEF, in which an optimal therapeutic approach has not yet been identified. Therefore, the gut microbiota and its metabolites may represent an alternative therapeutic target with which to approach HFpEF. 

The mechanisms leading to gut-microbiota-related changes in cardiac performance are still to be clarified. However, evidence shows that oxidative damage and inflammation after the modification of the gut microbiota seem to play a major role [87]. Subsequently, modulating inflammatory states [42] and counteracting oxidative damage due to metabolic imbalances appear to represent promising solutions to approaching the treatment of HFpEF. 

Recently, nutritional interventions have been found to significantly affect the composition of the gut microbiota [88]. In particular, many studies have shown that among patients undergoing HF, a diet characterised by a high fibre intake and short-chain fatty acids is associated with reduced gut dysbacteriosis [88]. Merques et al. demonstrated that a diet characterised by a low fibre intake is associated with negative cardiac remodelling that leads to hypertension and the progression of cardiac fibrosis [89]. Probiotics may represent an additional therapeutic strategy for preventing cardiovascular disease. In fact, it is known that probiotics are microorganisms whose ingestion and consequent interaction with gut flora may result in several benefits in terms of myocardial protection and inducing anti-inflammatory responses [90]. 

However, in this context, a limitation exists: most of the data obtained on the use of probiotics to treat CV disease, particularly HF, are derived from animal studies [91]. GutHeart is a multicentre prospective randomized open-label trial in which the authors randomized HF patients with a left ventricular ejection fraction (LVEF) <40% and NYHA class II-III to treatment (1:1:1) with Saccharomyces boulardii, the antibiotic rifaximin, or the standard of care only. The results at three months have not shown a difference in terms of LVEF, microbiota composition, laboratory biomarkers, or functional assessment [92].

Gut Microbiota enzymes, which promote negative CV effects, may constitute a therapeutic target. In particular, TMAO levels correlate with markers of inflammation, endothelial dysfunction in cases of HF, and diastolic impairment [93]. Therefore, targeting the molecular reactions leading to an increase in circulating TMAO, for example, the conversion of choline to TMAO, may be a valid plan of action with which to reduce the negative effects of HF and its comorbidities [94]. Recently, Steinke et al. stressed the importance of, for example, constructing DMB analogues to inhibit TMA formation via inhibiting CutC/D lyase or building a drug that mimics the structure of dietary indoles inhibiting FMO3 activity [95]. Finally, faecal transplantation may be a last-resort therapeutic strategy; however, no data are available regarding its application outside the treatment of Clostridium difficile infection [96].

## 9. Conclusions

It is becoming increasingly evident that the gut microbiota plays a role in the pathogenesis of many disorders, including CV disease and CHF. Therefore, a more detailed investigation of the link between the gut microbiota and the CV system is necessary. The common drugs that we routinely use for the treatment of HF have revolutionized the courses of patients with extremely compromising situations. Nevertheless, these drugs have proved to be insufficient to control the early stage of HF. Scientific evidence suggests that intervening in comorbidities and lifestyle modification is the only and most important way to impact the disease positively and precociously. In this context, therapeutic interventions concerning dysbiosis of the gut microbiota could represent a new and innovative instrument with which to improve the clinical course of HF. Further randomized clinical trials are required to define the correlation between gut microbiota and CHF and indicate the most appropriate therapeutic approach.

## Figures and Tables

**Figure 1 ijms-24-11971-f001:**
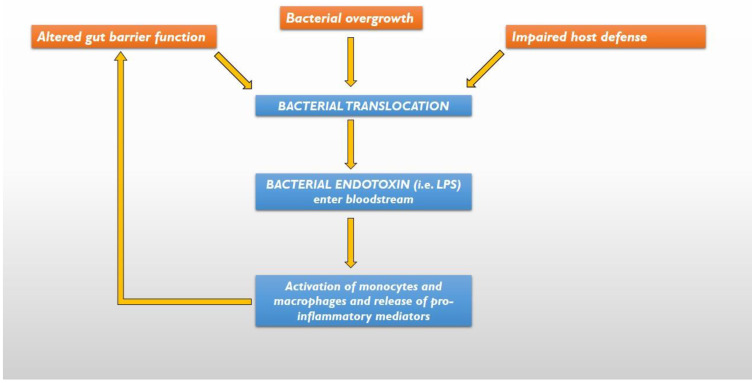
The 3 mechanisms responsible for bacterial translocation and systemic inflammation. LPS = lipopolysaccharide.

**Figure 2 ijms-24-11971-f002:**
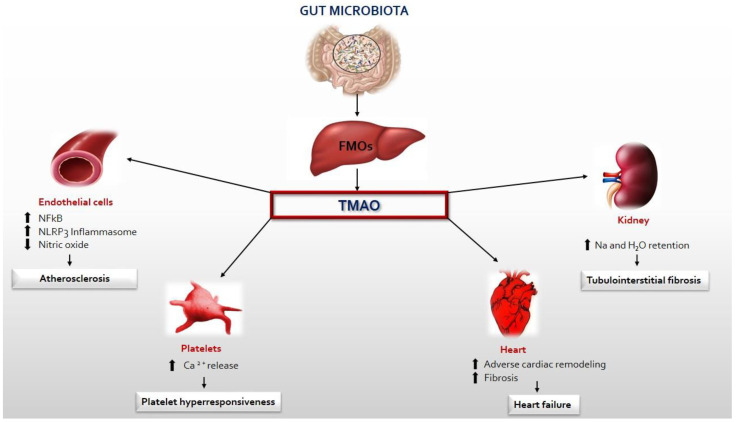
TMA produced by the gut microbiota enters the circulatory system and is oxidized to TMAO by FMO3 in the liver. Circulating TMAO causes endothelial dysfunction, platelet hyperreactivity, heart failure, and the progression of renal disease.

**Figure 3 ijms-24-11971-f003:**
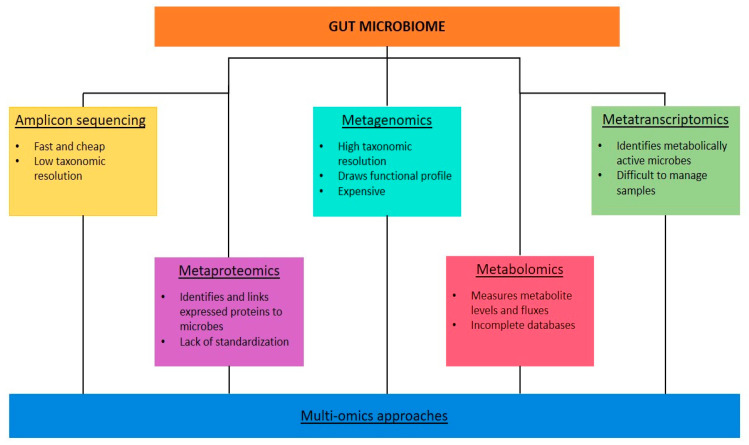
Different high-throughput-sequencing techniques performed at the level of microbes, DNA, mRNA, proteins, and metabolites.

## Data Availability

No new data were created or analysed in this study. Data sharing is not applicable to this article.

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
