# Peer review of "Gut Microbiota Composition and Cardiovascular Disease: A Potential New Therapeutic Target?"

_ijms, 2023, doi:10.3390/ijms241511971_

Round 1

Reviewer 1 Report

This review manuscript by Belli et al., merits to be published in IJMS as they review an important metabolite, trimethylamine-N-oxide (TMAO), which has been used as biomarker previously for a variety of diseases. It is only logical that it is linked to gut microbiota composition.

I have only minor points to report, only that perhaps the authors could compare and contrast with lipopolysaccharide a little more.

For example, is TMAO recognized by a immune receptor (e.g.,TLR)? \They state that (line 194-196) " TMAO promotes myocardial inflammation by increasing TNF-a levels and decreasing IL-10 levels leading to increased expression of pro-inflammatory cytokines and decreased expression of anti-inflammatory cytokines". In other words, what is the molecular receptor that initiates this cascade? Is it through co-signalling with LPS? If the upstream molecular mechanism is unknown, please elaborate to say only the downstream mechanism is only known. I thought it was only correlated with disease, not a cause. 

A second point that I find important to discuss is the rational drug discovery against the formation of TMAO. Recently, Steinke et al., 2020 (doi: 10.3389/fphys.2020.567899), discussed the efforts to design drugs against the bacterial TMA lyase CutC/D and the human FMO3 enzyme. I think this review would be well served to discuss these ongoing efforts.

Other formatting minor points: In Figure 1, please write "i.e., LPS" instead e.g. LPS bacterial endotoxin is actually lipid A portion of LPS, but most don't make that distinction.

Reviewer 2 Report

The current article under review highlights the interplay between the microbiota, its metabolites, and cardiovascular disease development and/or progression while emphasizing heart failure. The review aims at providing a direction for researchers to take this interaction into account while conducting future research. Overall, the manuscript will be of interest to the journal's readership and would serve as a guide for researchers to consider additional experimental designs to explore the relationship between the gut microbiome and cardiovascular diseases. The authors are requested to address the following comments.

1.      In the section talking about the link between gut microbiota and the heart, the authors have performed two different meta-analyses utilizing clinical trials. Please provide details on the inclusion and exclusion criteria of the clinical trials chosen. A flow chart involving what factors were considered to choose the specific clinical trials with the end goals for meta-analysis would be highly beneficial for the readers.

2.      Is there any significant role that reactive oxygen species may play as another marker and how might they affect this interaction?

3.      Please review and include the following references-

a.      Toya, Takumi, et al. "Coronary artery disease is associated with an altered gut microbiome composition." PloS one 15.1 (2020): e0227147.

b.      Liu, Honghong, et al. "Gut microbiota from coronary artery disease patients contributes to vascular dysfunction in mice by regulating bile acid metabolism and immune activation." Journal of translational medicine 18 (2020): 1-18.

Minor comments: Please include reference for line 113 and proofread the article for minor English language corrections (for instance, in line 243, the sentence starts with “Anyway” and does not sound scientific).

Please see the minor comments for improving the readability of the manuscript.
